# Reproducible diagnostic metabolites in plasma from typhoid fever patients in Asia and Africa

Elin Näsström[1], Christopher M Parry[2,3], Nga Tran Vu Thieu[4,5], Rapeephan R Maude[6], Hanna K de Jong[7,8], Masako Fukushima[2], Olena Rzhepishevska[1], Florian Marks[9], Ursula Panzner[9], Justin Im[9], Hyonjin Jeon[9], Seeun Park[9], Zabeen Chaudhury[9], Aniruddha Ghose[10], Rasheda Samad[10], Tan Trinh Van[4], Anders Johansson[11], Arjen M Dondorp[6], Guy E Thwaites[4,12], Abul Faiz[13], Henrik Antti[1]*, Stephen Baker[4,12,14]*

[1]Department of Chemistry, Computational Life Science Cluster, Umeå University, Umeå, Sweden; [2]Clinical Sciences, Liverpool School of Tropical Medicine, Liverpool, United Kingdom; [3]School of Tropical Medicine and Global Health, Nagasaki University, Nagasaki, Japan; [4]The Hospital for Tropical Diseases, Wellcome Trust Major Overseas Programme, Oxford University, Oxford, United Kingdom; [5]Clinical Research Unit, Ho Chi Minh City, Vietnam; [6]Mahidol-Oxford Tropical Medicine Research Unit (MORU), Faculty of Tropical Medicine, Mahidol University, Bangkok, Thailand; [7]Department of Internal Medicine, Division of Infectious Diseases and Center for Infection and Immunity Amsterdam (CINIMA), University of Amsterdam, Amsterdam, the Netherlands; [8]Center for Experimental Molecular Medicine (CEMM), Academic Medical Center, University of Amsterdam, Amsterdam, The Netherlands; [9]The International Vaccine Institute, Seoul, South Korea; [10]Chittagong Medical College Hospital, Chittagong, Bangladesh; [11]Department of Clinical Microbiology, Umeå University, Umeå, Sweden; [12]Centre for Tropical Medicine, Oxford University, Oxford, United Kingdom; [13]Malaria Research Group and Dev Care Foundation, Dhaka, Bangladesh; [14]Department of Medicine, The University of Cambridge, Cambridge, United Kingdom

*For correspondence: henrik.antti@umu.se (HA); sbaker@oucru.org (SB)

Competing interests: The authors declare that no competing interests exist.

**Abstract** *Salmonella* Typhi is the causative agent of typhoid. Typhoid is diagnosed by blood culture, a method that lacks sensitivity, portability and speed. We have previously shown that specific metabolomic profiles can be detected in the blood of typhoid patients from Nepal (Näsström et al., 2014). Here, we performed mass spectrometry on plasma from Bangladeshi and Senegalese patients with culture confirmed typhoid fever, clinically suspected typhoid, and other febrile diseases including malaria. After applying supervised pattern recognition modelling, we could significantly distinguish metabolite profiles in plasma from the culture confirmed typhoid patients. After comparing the direction of change and degree of multivariate significance, we identified 24 metabolites that were consistently up- or down regulated in a further Bangladeshi/Senegalese validation cohort, and the Nepali cohort from our previous work. We have identified and validated a metabolite panel that can distinguish typhoid from other febrile diseases, providing a new approach for typhoid diagnostics.

## Introduction

Typhoid is a systemic infection caused by the bacterium *Salmonella* Typhi (*S.* Typhi) (*Parry et al., 2002*; *Dougan and Baker, 2014*). With an estimated 21 million cases annually, typhoid remains a persistent global health issue (*Buckle et al., 2012*; *Ochiai et al., 2008*). The symptoms of typhoid arise after the organism invades the gastrointestinal wall and enters the bloodstream (*Everest et al., 2001*). Isolating the organism from the bloodstream is the mainstay of typhoid diagnostics (*Gilman et al., 1975*; *Parry et al., 2011*), but this method lacks sensitivity and researchers are aiming to discover biomarkers that may become a more reliable and rapid approach to diagnosing disease (*Baker et al., 2010*). One approach for discovering biomarkers is metabolomics, a method detecting low-molecular-weight metabolites in biological materials by mass spectrometry (*Madsen et al., 2010*). Our previous work demonstrated that significant and reproducible metabolite profiles could segregate *S.* Typhi cases, *Salmonella* Paratyphi A cases, and asymptomatic controls in a Nepali patient cohort (*Näsström et al., 2014*). Further, we found that a combination of six metabolites could define the infecting pathogen in the blood of febrile patients. These data represented a major step forward in the discovery of biomarkers with the potential to be future typhoid diagnostics. We have applied a similar approach with plasma samples collected from febrile patients in Bangladesh and Senegal to further investigate and validate our previous findings.

## Results

### Plasma metabolites in Bangladeshi typhoid fever patients

By hierarchical multivariate curve resolution, we resolved 394 peaks from the GCxGC-TOFMS data (Materials and methods) in 30 plasma samples from febrile patients in Bangladesh (*Table 1*); after filtering to remove low-quality peaks and metabolites with a high run order correlation we detected 236 metabolite peaks suitable for modeling. Of the detected metabolite peaks, 65/236 (27.5%) had a putative annotation, 8/236 (3.4%) had a metabolite class, 32/236 (13.6%) were of uncertain identity, and 131/236 (55.5%) were unknown (*Supplementary file 1A*). Initial modeling of these 236 metabolites revealed one outlying sample in the fever control group, which was excluded. We applied a supervised pattern recognition approach using Orthogonal Partial Least Squares with Discriminant Analysis (OPLS-DA) to differentiate the metabolite profiles between two sample classes (culture positive typhoid patients and fever controls). This model was then used to predict the identity of the individual samples in a third sample class (clinically suspected typhoid). The OPLS-DA model provided excellent predictive power for distinguishing between culture-positive typhoid patients and fever controls in the first predictive component using 236 informative primary metabolite features (t[1] and tcv[1]) (p=0.006) (*Figure 1A* and *Supplementary file 1B*).

### Prediction of culture-negative/clinically suspected typhoid fever

A major challenge in diagnosing typhoid is identifying true typhoid patients but have a negative blood culture result (*Moore et al., 2014*). We observed a significant overlap between the culture-negative/clinically suspected typhoid metabolite profiles with both the culture-positive group and the fever control group (*Figure 1B*). We used the OPLS-DA model that distinguished between the culture-positive typhoid patients and the fever controls to predict the clinically suspected typhoid samples. We found that 5/9 plasma samples had a metabolite profile indicative of culture-positive typhoid and three exhibited a greater degree of resemblance to fever controls (one indifferent) (*Figure 1B*). Notably, 3/5 clinically suspected typhoid samples with a metabolite profile indicative of typhoid were additionally PCR amplification positive for *S.* Typhi in blood (*Table 1* and *Figure 1B*). We also investigated potential diagnostic typhoid signatures in urine samples from the same patients using UPLC-Q-TOFMS (Materials and methods). Examination of 941 putative metabolite peaks obtained from urine using positive ionization an OPLS-DA model resulted in significantly different metabolite profiles between the *S.* Typhiculture-positive patients and the fever controls (p=0.025) (*Figure 1—figure supplement 1* and *Supplementary file 1B*).

**Table 1.** Patient group metadata for the Bangladeshi cohort.

| Clinical parameter* | Culture confirmed typhoid[†] (n = 10)[¶] | Suspected typhoid[‡] (n = 9) | Fever controls[§] (n = 10) |
|---|---|---|---|
| Age (years) | 23 (20–30) | 22 (16–30) | 46 (20–65) |
| Sex (male) | 5 | 6 | 8 |
| Fever duration (days) | 7 (5–11) | 10 (6–12) | 5 (5–9) |
| Abdominal pain | 5 | 5 | 5 |
| Diarrhoea | 5 | 3 | 2 |
| Constipation | 1 | 4 | 2 |
| Vomiting | 5 | 7 | 3 |
| Cough | 2 | 1 | 7 |
| Rash | 0 | 2 | 0 |
| Dysuria | 0 | 0 | 2 |
| Headache | 6 | 4 | 6 |
| Seizure | 0 | 0 | 0 |
| Drowsy | 0 | 2 | 2 |
| Bloody stool/ Melaena | 1 | 0 | 1 |
| Confusion/unconscious | 0 | 1 | 0 |
| Axillary temperature (°C) | 38.6 (38.3–39.4) | 38.9 (38.5–39.4) | 38.6 (38.3–38.9) |
| Pulse (bpm) | 108 (97–114) | 100 (92–110) | 105 (86–123) |
| Jaundice | 0 | 1 | 1 |
| Hepatomegaly | 1 | 2 | 1 |
| Splenomegaly | 0 | 0 | 1 |
| Hb | 11.9 (10.4–12.9) | 12.0 (9.6–12.1) | 12.6 (10.4–14.0) |
| WBC | 6.6 (4.9–8.4) | 7.0 (4.2–8.5) | 14.4 (10.7–21.1) |
| Neutrophils (%) | 79 (70–81) | 70 (63–72) | 80 (78–88) |
| Lymphocytes (%) | 19 (15–26) | 25 (24–33) | 15 (8–18) |
| Monocytes (%) | 2 (2–3) | 2 (2–4) | 2 (2–4) |
| Eosinophils (%) | 1 (1–1) | 1 (1–2) | 1 (1–2) |
| Platelets | 170 (160–232) | 180 (160–265) | 280 (180–320) |
| Urea | 24.6 (21.4–28.0) | 24.1 (22.1–29.5) | 67.9 (21.4–81.2) |
| Creatinine | 0.9 0.6–1.0) | 0.9 (.7–1.0) | 1.6 (0.8–2.4) |
| AST | 101 (47–137) | 51 (33–199) | 32 (16–78) |
| ALT | 93 (48–137) | 36 (28–105) | 31 (20–43) |
| Complications** | 2 | 0 | 4 |
| Died[††] | 0 | 0 | 2 |

*Median and interquartile range (IQR) for each patient group given for quantitative parameters and number of patients with presence of symptom/characteristics for qualitative parameters.

[†]Typhoid confirmed by a positive blood culture for *S*. Typhi.

[‡]Clinical suspected typhoid fever with a negative blood culture, final diagnoses included: Clinically suspected typhoid with blood PCR amplification positive for *S*. Typhi (3); clinical suspected typhoid (4); clinical suspected typhoid or leptospirosis (1); possible typhoid encephalopathy (1).

[§]Fever controls included: pneumonia (3), malaria (2), meningitis (2); sepsis (1), cellulitis (1), urinary tract infection (1).

[¶]One sample removed from analysis due to discrepant metabolite profile.

**Complications were: gastrointestinal bleeding and severe anaemia requiring transfusion in the typhoid group; respiratory failure, hepatorenal failure, septic shock and cardiopulmonary arrest, coma and cardiopulmonary arrest, and an acute myocardial infarction in the fever controls.

[††]Deaths in this group were associated with sepsis and malaria.

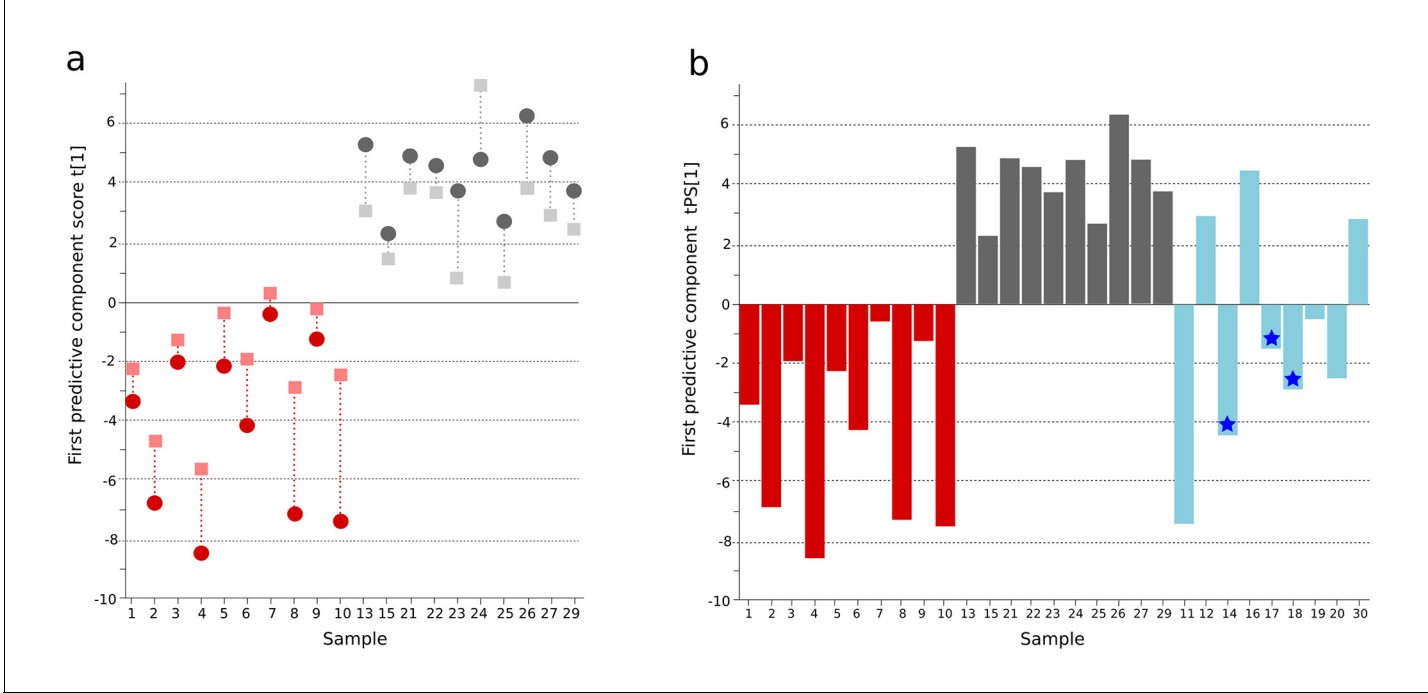

**Figure 1.** OPLS-DA model of plasma metabolites from a Bangladeshi cohort of patients with culture-positive typhoid and fever controls, with prediction of suspected typhoid. (**A**) OPLS-DA model generated from GCxGC-TOFMS data from the plasma of 10 patients with culture-positive typhoid and 10 fever controls using 236 metabolites. Regular (circles) and cross-validated (squares) scores for the first predictive component (t[1] and tcv[1], respectively, linked by broken line) showing a separation between culture-positive typhoid (red) and fever control samples (grey) (p=0.006). (**B**) Column plot of the predicted scores for the first predictive component (tPS[1]) where clinically suspected typhoid samples (n = 9) (blue columns) have been predicted into the model distinguishing between culture-positive typhoid (red) and fever control samples (grey). Plot shows five samples were more similar to the culture-positive typhoid samples and three more similar to the controls; one sample remained marginal. The blue stars identify PCR-amplification-positive samples.

The following figure supplement is available for figure 1:

**Figure supplement 1.** OPLS-DA model of urine metabolites from a Bangladeshi cohort including patients with culture-positive typhoid and fever controls, with prediction of suspected typhoid.

## Reproducible typhoid metabolite patterns in Bangladeshi and Nepali cohorts

We next compared informative plasma metabolites of Bangladeshi *S.* Typhiculture-positive patients with the metabolites in the *S.* Typhi patients from our previous investigation in Nepal (*Näsström et al., 2014*). We found 99 informative metabolites in plasma from both cohorts. Comparing the direction of change and the degree of significance we identified 33 metabolites that were consistently up- or downregulated between the culture-positive *S.* Typhi patients and fever/asymptomatic controls in the two studies (*Supplementary file 1C*). Fifteen of the 33 metabolites were multivariate significant with a stricter criteria ($w^*>|\bar{x} \pm SD|$) in the Bangladeshi cohort and all 33 metabolites were multivariate significant ($w^*>|0.03|$) in the Nepali cohort. OPLS-DA models with the 15 multivariate significant metabolites resulted in significant separations between *S.* Typhiculture-positive patients and fever controls in the current study (Bangladeshi cohort) (p=0.016), and the asymptomatic controls in the previous study (Nepali cohort) (p<0.0001) (*Figure 2* and *Supplementary file 1B*). Models based on all 33 correspondingly up or downregulated metabolites could also distinguish the *S.* Typhiculture-positive patients from the fever/asymptomatic controls (current study: p=0.077, previous study: p<0.0001) (*Supplementary file 1B*).

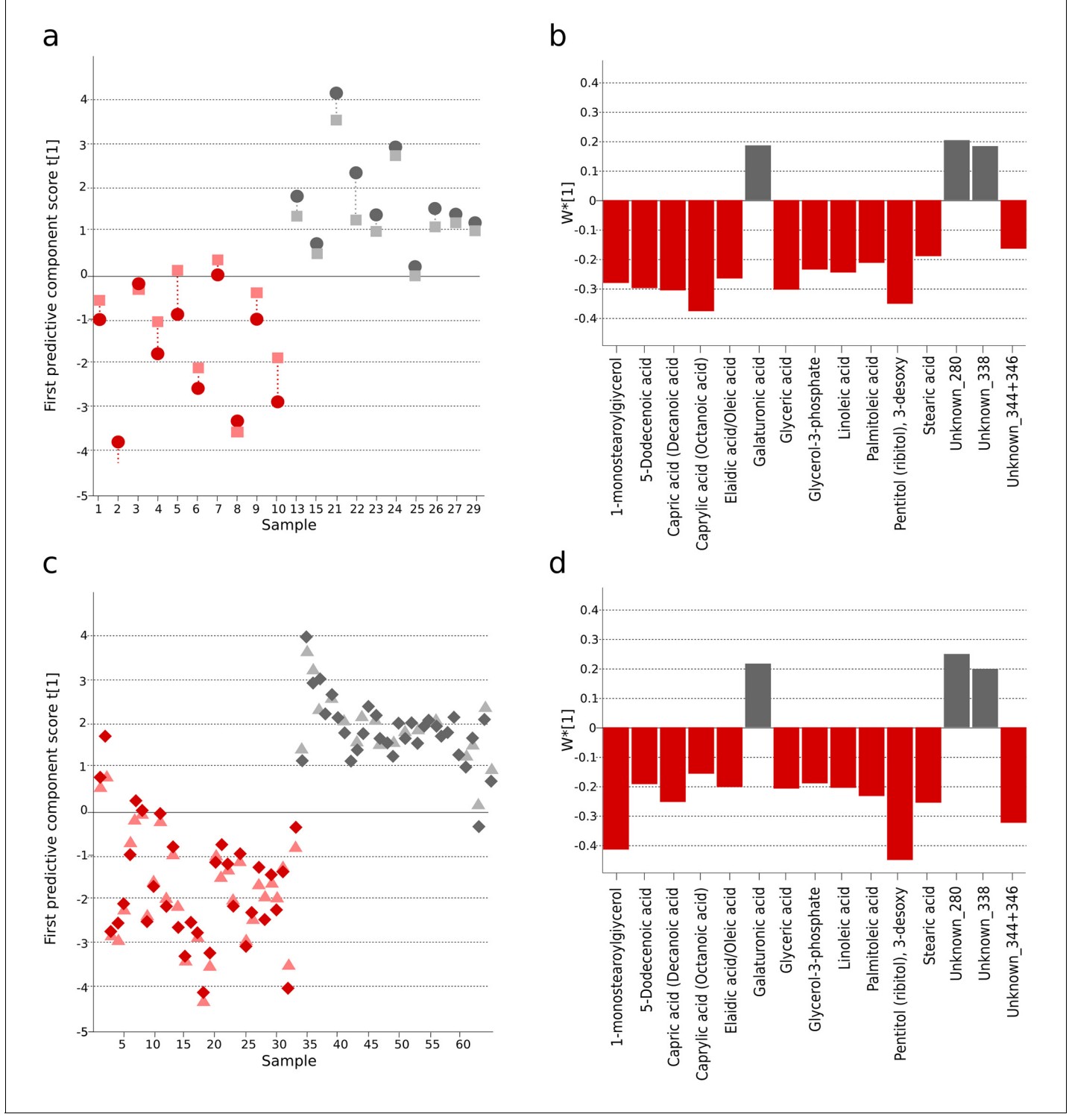

**Figure 2.** The identification and validation of typhoid diagnostic metabolites. OPLS-DA models generated from GCxGC-TOFMS data using 15 informative metabolites from the current study (Bangladeshi cohort) and the previous study in Nepali cohort that were consistently up- or downregulated and significantly different in a multivariate model separating culture-positive *S*. Typhi patients from controls. (**A**) Regular (circles) and cross-validated (squares) scores for the first predictive component (t[1] and tcv[1], respectively, linked by broken line) showing a separation between culture-positive typhoid (red; n = 10) and fever control samples (grey; n = 10) (p=0.016) in the Bangladeshi cohort. (**B**) Column plot of model covariance loadings (w*[1]) for the first predictive component for the 15 common named metabolites in the Bangladeshi cohort, showing metabolites with a higher relative concentration in the culture-positive typhoid group in red and metabolites with a higher relative concentration in the fever control group in grey. (**C**) Regular (circles) and cross-validated (squares) scores for the first predictive component (t[1] and tcv[1], respectively, linked by broken line)

*Figure 2 continued on next page*

*Figure 2 continued*

showing a separation between culture-positive typhoid (red; n = 33 including eight analytical replicates) and afebrile control samples (grey; n = 32 including seven analytical replicates) (p<0.0001) from the Nepali cohort. (**D**) Column plot of model covariance loadings (w*[1]) for the first predictive component for the 15 common named metabolites in the Nepali cohort, showing metabolites with a higher relative concentration in the typhoid group in red and metabolites with a higher relative concentration in the afebrile control group in grey.

## Typhoid fever metabolites in Bangladeshi and Senegalese validation cohorts

For further validation, we analyzed an additional 54 plasma samples from febrile patients from Bangladesh and Senegal using a different analytical technique (GC-TOFMS, methods). This validation cohort included samples from patients with confirmed typhoid and samples from patients with malaria or infections caused by other pathogens. Through an independent targeted processing approach, we detected 247 putative metabolites; after manual filtering, 104 metabolites were suitable for modeling (*Supplementary file 1D*). Initially, a three-class OPLS-DA model was obtained indicating the discrimination of typhoid samples from the two control groups (malaria and other pathogens) (*Figure 3—figure supplement 1* and *Supplementary file 1B*). Furthermore, a two-class OPLS-DA model for separation between typhoid and all control samples together showed significant separation for the new Bangladeshi samples (one overlapping control) and the majority of the Senegalese samples (p<0.0001) (*Figure 3A* and *Supplementary file 1B*). Malaria presents with a clinical syndrome that can be indistinguishable from typhoid fever; therefore, distinguishing between the diseases using their metabolite profiles is an important diagnostic approach. The typhoid samples were compared to the malaria positive samples in a separate OPLS-DA model and showed significant separation (p=0.0001), with two overlapping Senegalese typhoid samples, potentially signifying co-infection (*Figure 3B* and *Supplementary file 1B*).

The informative plasma metabolites from the Bangladeshi/Senegalese validation cohort were compared to the primary Bangladeshi and Nepali cohorts. We identified 49 common metabolites across all datasets. After comparing the direction of change and degree of multivariate significance, we found 24 metabolites that were consistently up- or downregulated in the Bangladeshi/Senegalese validation cohort and the Bangladeshi cohort and/or the Nepali cohort (*Supplementary file 1D*). OPLS-DA models of the consistently up- or downregulated metabolites resulted in significant separations between those with typhoid and the control samples for the Bangladeshi/Senegalese validation cohort (p<0.0001) (*Figure 3—figure supplement 2A*) and for the Nepali cohort (p<0.0001) (*Figure 3—figure supplement 2C*), the model was weaker for the primary Bangladeshi cohort (p=0.39) (*Figure 3—figure supplement 2B*) (*Supplementary file 1B*).

## Discussion

This study augments our previous findings and provides additional insight into next generation typhoid diagnostics (*Näsström et al., 2014*). Previously, we aimed to identify metabolite profiles that could distinguish between patients with *S.* Typhi and *S.* Paratyphi A infections. We hypothesized that metabolite profiles might differentiate clinically indistinguishable infections caused by these genetically related pathogens (*Didelot et al., 2007*; *Maskey et al., 2006*); asymptomatic individuals constituted the control group. Here, we aimed to identify *S.* Typhi metabolite profiles in different settings without *S.* Paratyphi A disease (*Maude et al., 2015*). This approach was a greater challenge given a heterogeneous fever control group and a group of patients with suspected typhoid fever. We suggest this study more closely reflects a real situation given the non-specific presentation of febrile diseases. We also assessed the diagnostic potential of urine using this methodology as it is a convenient specimen (*Gilman et al., 1975*).

Using a validation cohort from Asia and Africa we were able to identify significant, reproducible metabolite profiles in the blood of patients with typhoid. The identified metabolites significantly discriminated *S.* Typhi-culture-positive individuals from patients with alternative febrile diseases, including malaria. Among patients with clinically suspected typhoid but a negative blood culture, we identified metabolite profiles consistent with the confirmed typhoid patient profiles (*Nga et al., 2010*). The metabolite profiles in urine also significantly segregated the typhoid patients from the

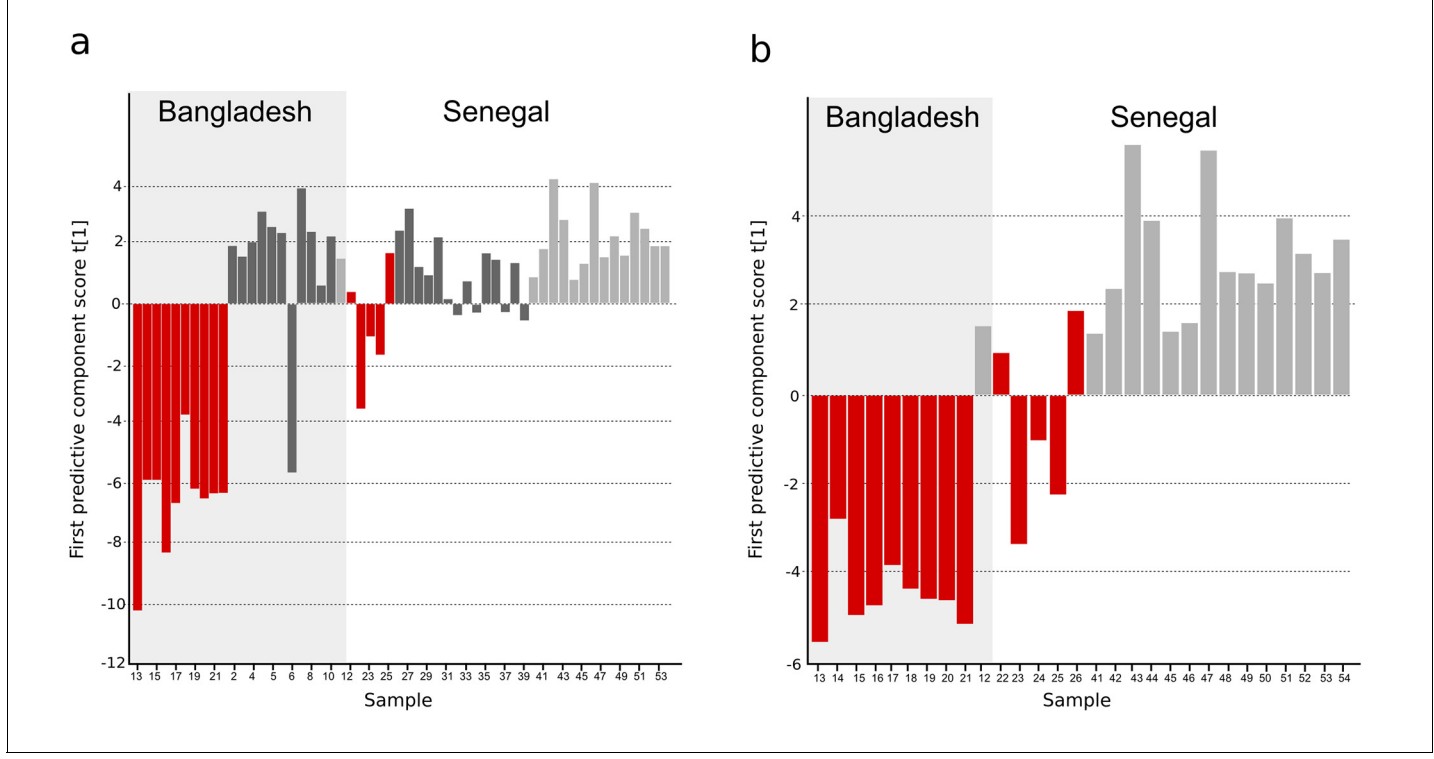

**Figure 3.** OPLS-DA models of plasma metabolites from Bangladeshi/Senegalese validation cohorts of patients with typhoid, malaria and other infections. OPLS-DA models generated from GC-TOFMS using 104 metabolites. (A) Column plot of the first predictive component scores, t[1] showing a separation of typhoid infection samples (red; n = 14) from the two control groups; malaria (light grey; n = 15) and infections caused by other bacteria/pathogens (grey; n = 25) (p<0.0001). For the Bangladeshi samples, there is a clear separation except for one control sample behaving as a typhoid sample, there is more overlap for the Senegalese samples. (B) Column plot of the first predictive component scores, t[1] showing a separation of typhoid infection samples (red; n = 14) from malaria samples (light grey; n = 15) (p<0.001). There is a clear separation for the Bangladeshi samples and for the Senegalese samples except two typhoid samples behaving as malaria.

The following figure supplements are available for figure 3:

**Figure supplement 1.** Three-class OPLS-DA model of GC-TOFMS data of plasma samples from a Bangladeshi/Senegalese validation cohort including patients with typhoid, malaria and other infections based on 104 metabolites.

**Figure supplement 2.** Comparison of metabolites for three sample cohorts.

febrile controls, but did not provide the same predictions as the plasma samples for the culture-negative patients. The culturenegative clinically suspected typhoid group is challenging because of the lack of a satisfactory reference standard diagnostic test, but this innovative method allows a new approach to investigate this problematic patient group using plasma samples.

The most important finding from this study was the identification and validation of significantly variable metabolites that can identify blood culture confirmed typhoid fever patients in distinct patient cohorts (Asia and Africa) with differing control populations. At least 24 metabolites have the potential to identify typhoid fever patients in these patients. These included glycerol-3-phosphate (carbon source and precursor for phospholipid biosynthesis) (*Austin and Larson, 1991*), stearic acid (component of liposome)(*Galdiero et al., 1994*), and linoleic acid (bactericidal activity) (*Yang et al., 2010*), pyruvic acid, and creatinine. Furthermore, leucine and phenylalanine were consistently up- or downregulated between all collections.

New approaches are needed for the diagnosis of tropical febrile diseases. We have identified and validated a panel of metabolites that can identify febrile patients with typhoid. The next challenges are to corroborate these targets in larger patient numbers and incorporate into simple diagnostic test formats. This approach could be potentially expanded into other tropical febrile diseases.

## Materials and methods

To measure the systemic metabolite profiles associated with typhoid, we selected plasma and urine samples from 30 patients in a febrile disease study conducted in Chittagong, Bangladesh (*Maude et al., 2015*): Ten patients had blood culture *S.* Typhi confirmed typhoid; nine patients had a clinical diagnosis of typhoid (blood culture negative ± PCR positive for *S.* Typhi); and 11 matched individuals had a febrile disease other than typhoid (fever controls) (*Table 1* and *Supplementary file 2*). The study sites, population and study design are described in detail in the supplementary information and are published elsewhere (*Maude et al., 2015*). Validation samples included plasma samples from 54 patients from Bangladesh and Senegal with 14 patients having confirmed *S.* Typhi infection, 15 patients having malaria and 25 having an infection caused by other bacteria/pathogens (*Supplementary file 2*) (*von Kalckreuth et al., 2016*; *Marks et al., 2017*). Chromatograms and mass spectra of the Bangladeshi plasma samples were generated and analysed as previously described by blinded operator in a random order using comprehensive two-dimensional gas chromatography with time-of-Flight Mass Spectrometry (GCxGC-TOFMS) (*Näsström et al., 2014*). Chromatograms and mass spectra of urine samples were generated using high-throughput ultra-performance liquid chromatography/quadrupole-time-of-flight mass spectrometry (UPLC-Q-TOFMS). Chromatograms and mass spectra of the Bangladeshi/Senegalese validation plasma samples were generated using one-dimensional gas chromatography with time-of-flight mass spectrometry (GC-TOFMS). Acquired and processed data was analyzed using chemometrics based pattern recognition. All methods are described in detail in *Supplementary file 3*.

## Acknowledgements

This project was funded by the Wellcome Trust of Great Britain (106158/Z/14/Z). Stephen Baker is a Sir Henry Dale Fellow, jointly funded by the Wellcome Trust and the Royal Society (100087/Z/12/Z). Henrik Antti is funded by the Swedish Research Council (VR-U 2015–03442).

## Additional information

### Funding

| Funder | Grant reference number | Author |
|---|---|---|
| Vetenskapsrådet | VR-NT 2010-4284 | Elin Näsström Henrik Antti |
| Swedish Research Council | VR-U 2015-03442 | Henrik Antti |
| Wellcome | 106158/Z/14/Z | Stephen Baker |
| Wellcome | 100087/Z/12/Z | Stephen Baker |

The funders had no role in study design, data collection and interpretation, or the decision to submit the work for publication.

### Author contributions

EN, Conceptualization, Resources, Data curation, Formal analysis, Methodology, Writing—original draft, Writing—review and editing; CMP, Conceptualization, Investigation, Methodology, Project administration; NTVT, Data curation, Funding acquisition, Investigation, Methodology; RRM, HKdJ, MF, ZC, AG, RS, TTV, AMD, AF, Investigation, Methodology; OR, Formal analysis, Visualization, Methodology; FM, Resources, Supervision, Project administration; UP, Resources, Data curation; JI, HJ, SP, Resources, Investigation, Methodology; AJ, Investigation, Methodology, Writing—review and editing; GET, Supervision, Investigation, Methodology; HA, Conceptualization, Data curation, Formal analysis, Supervision, Funding acquisition, Investigation, Writing—original draft, Writing—review and editing; SB, Conceptualization, Formal analysis, Supervision, Funding acquisition, Methodology, Writing—original draft, Project administration, Writing—review and editing

### Author ORCIDs

Stephen Baker, http://orcid.org/0000-0003-1308-5755

## Ethics

Human subjects: The study was conducted according to the principles expressed in the Declaration of Helsinki. The Bangladesh National Research Ethical Committee (BMRC/NREC/2010-2013/1543), the Chittagong Medical College Hospital Ethical Committee, the Oxford Tropical Research Ethics Committee (OXTREC 53-09), the Research Ethics committee of the Liverpool School of Tropical Medicine and Institute Pasteur de Dakar, Senegal, and the International Vaccine Institute, Republic of South Korea gave ethical approval for the study. Informed written or thumbprint consent was taken from the subject, their parent or caretaker for all enrolees.

## Additional files

### Supplementary files

• Supplementary file 1. (A) Table of detected metabolites in plasma samples analyzed with GCxGC-TOFMS in the primary Bangladeshi cohort. (B) Overview of multivariate models. (C) Table of common metabolites between the Bangladeshi and the previous Nepali cohort using OPLS-DA models of culture-positive typhoid infection vs. control. (D) Table of detected metabolites in plasma samples analyzed with GC-TOFMS in the Bangladeshi/Senegalese validation cohort.

• Supplementary file 2. Additional patient metadata and diagnoses.

• Supplementary file 3. Additional materials and methods.

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
