## [Decision Letter]

Thank you for submitting your article "Reproducible diagnostic metabolites detected in the plasma and urine of typhoid fever patients" for consideration by *eLife*. Your article has been reviewed by two peer reviewers, and the evaluation has been overseen by a Reviewing Editor and Prabhat Jha as the Senior Editor. The reviewers have opted to remain anonymous.

The reviewers have discussed the reviews with one another and the Senior Editor has drafted this decision to help you prepare a revised submission.

Summary:

This aim of this study was to identify *Salmonella* Typhi specific metabolites in plasma and urine samples in order to develop a more sensitive diagnostic test for typhoid fever. The samples were from three groups of Bangladeshi patients: confirmed typhoid cases (n=10), suspected typhoid cases (n=9) and fever controls (n=10) and were analyzed by two-dimensional gas chromatography with time-of-flight mass spectrometry (plasma) or high throughput ultra-performance liquid chromatography/quadrupole time-of-flight mass spectrometry (urine).

This study follows up a previous one that identified serovar-specific plasma metabolites after analysis of 3 patient groups (*Salmonella* Typhi confirmed cases, n=25, *Salmonella* Paratyphi confirmed cases, n=25; afebrile controls, n=25) in Nepal. The data presented in this manuscript complements findings previously published by the team on enhancing diagnosis and surveillance of S typhi in febrile patients through a panel of metabolites. These data are very promising but the conclusions are premature given: i) the small number of samples tested, and ii) lack of information on the febrile controls – more information needs to be provided than the fact that they had other febrile diseases.

Essential revisions:

These promising findings could be significantly strengthened through validation of the identified metabolite pattern with a larger sample size. One suggestion is to use samples collected during routine surveillance from different countries to see how predictive these metabolites are in identifying typhoid. Additionally, it will be important to include controls with infections often confused with typhoid fever such as malaria or bacteremia/septicemia caused by *Enterobacteriaceae*, in particular non-typhoidal *Salmonella* such as Typhimurium or Enteritidis (which share a same O antigen).

---

## [Author Response]

*Summary:*

*This aim of this study was to identify Salmonella Typhi specific metabolites in plasma and urine samples in order to develop a more sensitive diagnostic test for typhoid fever. The samples were from three groups of Bangladeshi patients: confirmed typhoid cases (n=10), suspected typhoid cases (n=9) and fever controls (n=10) and were analyzed by two-dimensional gas chromatography with time-of-flight mass spectrometry (plasma) or high throughput ultra-performance liquid chromatography/quadrupole time-of-flight mass spectrometry (urine).*

*This study follows up a previous one that identified serovar-specific plasma metabolites after analysis of 3 patient groups (Salmonella Typhi confirmed cases, n=25, Salmonella Paratyphi confirmed cases, n=25; afebrile controls, n=25) in Nepal. The data presented in this manuscript complements findings previously published by the team on enhancing diagnosis and surveillance of S typhi in febrile patients through a panel of metabolites. These data are very promising but the conclusions are premature given: i) the small number of samples tested, andii) lack of information on the febrile controls – more information needs to be provided than the fact that they had other febrile diseases.*

*Essential revisions:*

*These promising findings could be significantly strengthened through validation of the identified metabolite pattern with a larger sample size. One suggestion is to use samples collected during routine surveillance from different countries to see how predictive these metabolites are in identifying typhoid. Additionally, it will be important to include controls with infections often confused with typhoid fever such as malaria or bacteremia/septicemia caused by Enterobacteriaceae, in particular non-typhoidal Salmonella such as Typhimurium or Enteritidis (which share a same O antigen).*

In response to the comments of the reviewers we have analysed an additional dataset including patient samples from both Bangladesh and Senegal. This validation cohort includes samples from patients with culture confirmed typhoid, patients with infection caused by other bacteria/pathogens and also a group of patients with confirmed malaria.

Changes made to the resubmitted manuscript

Title

Previous title: Reproducible diagnostic metabolites detected in the plasma and urine of typhoid fever patients

New title: Reproducible diagnostic metabolites in plasma from typhoid fever patients in Asia and Africa

Added authors

Authors associated with the Bangladeshi/Senegalese validation cohort were added

Abstract

Modification of the Abstract due to addition of Bangladeshi/Senegalese validation cohort

Introduction

Minor modification due to addition of Bangladeshi/Senegalese validation cohort

Results

Minor modifications of text in the two first paragraphs about the results of the primary Bangladeshi cohort

Removed details about urine analysis of the primary Bangladeshi cohort and moved figure to supplementary

Removed details in section about common metabolites between previous Nepali cohort and current primary Bangladeshi cohort

Added results from the Bangladeshi/Senegalese validation cohort. The results include multivariate models of all metabolites comparing typhoid samples to first all control samples then to malaria only. Metabolites from the Bangladeshi/Senegalese validation cohort were compared to the primary Bangladeshi cohort and to the previous Nepali cohort and a new set of 24 metabolites were outlined as being consistently up or down regulated in the Bangladeshi/Senegalese validation cohort and the primary Bangladeshi cohort and/or the previous Nepali cohort. Multivariate models of all three cohorts were made with these consistently regulated metabolites

Figures

Figure 1 with primary Bangladeshi cohort results remains

Moved figure with result of urine analysis of the primary Bangladeshi cohort to supplementary figure (previously Figure 2, now Figure 1—figure supplement 1)

Changed number of figure with common metabolites between previous Nepali cohort and current primary Bangladeshi cohort (previously Figure 3, now Figure 2)

Added figure of Bangladeshi/Senegalese cohort (Figure 3)

Added supplementary figure of Bangladeshi/Senegalese cohort, 3-class overview model (Figure 3—figure supplement 1)

Added supplementary figure of all three cohorts with metabolites consistently regulated in the Bangladeshi/Senegalese validation cohort and the primary Bangladeshi cohort and/or the previous Nepali cohort (Figure 3—figure supplement 2)

Tables

Table 1 with patient metadata for the primary Bangladeshi cohort remains with minor modifications

Discussion

The Bangladeshi/Senegalese validation cohort resulted in further validation of the significant metabolites differentiating culture confirmed typhoid from febrile controls (including malaria) in different patient cohorts from both Asia and Africa

Change in discussion of metabolites due to the addition of the Bangladeshi/Senegalese validation cohort

Methods

Added short description of methods for the Bangladeshi/Senegalese validation cohort

Supplementary information

Updated supplementary file with multivariate model overview ([Supplementary-material SD2-data])

Minor changes in supplementary files with metabolite lists. Detected metabolites for primary Bangladeshi cohort ([Supplementary-material SD1-data]) and common metabolites between previous Nepali cohort and current primary Bangladeshi cohort ([Supplementary-material SD3-data])

Changed number of supplementary file with additional Materials and methods (previously Supplementary file 4, now Supplementary file 6). Also, updated with methods for Bangladeshi/Senegalese validation cohort

Added supplementary file with detected metabolites for the Bangladeshi/Senegalese validation cohort (Supplementary file 5)

Added supplementary file with patient metadata and diagnosesfor the primary Bangladeshi cohort and for the Bangladeshi/Senegalese validation cohort, as requested by the reviewers

References

Change in a few references due to changed metabolites in the Discussion

All authors fulfil the criteria given in the authorship paragraph and no writing assistance was provided in the preparation of this manuscript. Our findings add new insights into typhoid diagnostics and will provide a springboard for future work into other unconventional diagnostic approaches. We think that this manuscript meets the stringent criteria of *eLife* and will be of considerable interest to your readership.